# Psychosocial Care Needs, Personality Styles, and Coping Strategies of Mourners in a Rural Municipality in Spain: An Observational Study

**DOI:** 10.3390/healthcare11091244

**Published:** 2023-04-27

**Authors:** Pedro-Ruymán Brito-Brito, Irayma Galdona-Luis, Martín Rodríguez-Álvaro, Alfonso-Miguel García-Hernández

**Affiliations:** 1Training and Research in Care, Primary Care Management Board of Tenerife, The Canary Islands Health Service, 38004 Santa Cruz de Tenerife, Spain; 2Department of Nursing, Faculty of Healthcare Sciences, University of La Laguna, 38200 Santa Cruz de Tenerife, Spain; 3Electronic Health Record Training, Primary Care Management Board of Tenerife, The Canary Islands Health Service, 38004 Santa Cruz de Tenerife, Spain; 4Santa Cruz de La Palma Primary Health Care Centre, Health Area of La Palma, The Canary Islands Health Service, 38200 Santa Cruz de Tenerife, Spain

**Keywords:** needs assessment, nursing diagnosis, grief, adaptation, psychological, personality

## Abstract

Grieving is a natural, self-limiting process of adaptation to a new reality following a significant loss, either real or perceived, with a wide range of manifestations that have an impact on the health of the grieving individual. This study aims to analyse the relationships between interpersonal styles, coping strategies, and psychosocial care needs in a sample of mourners in a rural municipality. Initial hypothesis: there are associations between types of grief and psychosocial needs, as well as between types of grief and interpersonal styles or coping strategies. An observational, descriptive, analytical, cross-sectional study was carried out with a sample of 123 people. Female participants represented 64.2% of all participants. The mean age was 42.7 (±13.2) years, and 86.2% of participants reported continuing to suffer from the loss, with a 10.5% prevalence of *maladaptive grieving*. Regarding the associations identified between coping strategies and the interpersonal characteristics of the mourners, we found that those with the best coping scores described themselves as self-confident, boastful, jovial, forceful, gentle-hearted, self-assured, outgoing, and/or neighbourly. By contrast, mourners who obtained poorer coping scores self-identified as shy, unsparkling, timid, unsociable, unbold, and/or bashful. This provides a clinical profile linked to *maladaptive grieving* in which emotional, self-perception, and social problems are prevalent.

## 1. Introduction

Grieving can be described as a natural, self-limiting process of adaptation to a new reality following a significant loss, either real or perceived, with a wide range of manifestations that have an impact on the health of the grieving individual [1]. Individuals respond to the death of a loved one in various ways, the most common of which include shock, disbelief, denial, high levels of anxiety, anger, sadness, and loss of sleep and appetite. Grief is not only related to the loss of a family member or a loved one, as different processes have been described that also lead to experiencing grief [2]. Examples include the loss of a job, the end of a friendship or relationship, life events that bring about changes, loss of functional ability to carry out daily activities, etc.

Grieving is a universal, unique, and painful human experience. In fact, its connotation of ‘sadness’ alludes to a family of constructs such as sorrow, mourning, and bereavement [3]. The emotional experience of coping with loss is referred to as the grieving process, which causes a need to adapt to a new situation [4]. In recent years, this process has received increasing attention in an attempt to better understand the development of normal or adaptive grieving, as it is a natural and necessary process through which ties with the deceased are maintained [5].

From a professional point of view, it is necessary to identify cases with potential risk for maladaptive grieving, which could result in prolonged pain, discomfort, and/or the occurrence of clinical conditions such as depression, panic disorders, and even psychotic breaks [6]. Therefore, the role of Family and Community Care nurses is key in assessing and identifying risky situations. By taking an emotional resilience-based approach, these nurses can provide adequate care to mourners [7]. It is well known that the provision of proper care at this level can assist in working towards a targeted grieving process. The nursing care process, together with standardised nursing languages, facilitates the integration of a theoretical framework for problem identification, planning of outcomes to be achieved, and the necessary care interventions [8].

Each individual experiences grief in a different way, facing a biography loaded with narrative, sensorial, experiential, biological, and behavioural meaning, which are not isolated from the clearly constantly changing socio-cultural trends in today’s highly variable society [9]. This is why interpersonal styles and coping strategies are particularly important in the context of grieving. In the field of health care in Spain, there are validated instruments to assess both interpersonal style and coping strategies [10,11], as well as tools to assess psychosocial care needs using standardised nursing language [12].

All this makes it possible to approach loss and grief from a constructivist and holistic point of view. It was in the late seventies and early eighties of the 20th century that Engel criticised the biomedical model—which, until then, had viewed health problems as physical and organic issues—and called for a more holistic approach that would integrate bio-, psycho-, and social spheres [13]. From this moment on, the proposal was that individuals should be assessed as a bio-psycho-social being, seeking to modify the previously established biomedical paradigm.

The diagnosis of grieving within the NANDA-I classification of nursing diagnoses was approved in 1980 and its definition and content have undergone changes, resulting in a shift in the understanding of grieving from a traditional perspective to a constructivist approach [14]. The latest NANDA-I classification [15] includes the following diagnoses: *readiness for enhanced grieving* (00285), *risk for maladaptive grieving* (00302), and *maladaptive grieving* (00301). As such, we believe that further insight into different types of grief could be obtained by describing the interpersonal characteristics and coping styles of grieving people living in a rural setting while taking their individual psychosocial needs into consideration. It is well known that rural settings provide a different environment than urban settings for support at the end of life, at the time of loss, and while grieving [16].

In light of the above, the primary objective of this study was to describe the relationships between interpersonal style, coping strategies, and psychosocial care needs among mourners in a rural municipality on the island of Tenerife (Canary Islands, Spain).

The initial hypothesis was the existence of associations between types of grief and psychosocial needs, as well as between types of grief and interpersonal styles or coping strategies.

## 2. Materials and Methods

### 2.1. Design and Sampling Method

This is an observational, descriptive, cross-sectional, analytical study reported according to the STROBE guidelines for the reporting of observational studies [17].

### 2.2. Study Setting

The study involves grieving people from the municipality of Güímar, Tenerife (Canary Islands, Spain). It is a largely rural municipality with a total population of 21,000. Between 2014–2019, a mean of 165 people/year died in the municipality [18]. The rural nature of the municipality is mainly due to its terrain and agricultural and fishing activities. As a result, many residents have adopted these activities as their main occupation.

Participants were recruited by including all those residents in the municipality of Güímar aged 18 or over who, after being briefed and signing the informed consent form explaining the study and its purpose, answered yes to at least one of the following two questions: (1) *Have you lost a loved one in the past year*? and (2) *Are you suffering from the loss of a loved one*? The exclusion criteria were having a cognitive impairment, a severe mental health problem, and/or sensory-perceptual, communication, or emotional difficulties that prevented the person from answering the questions in the field notebook. A non-probability, purposive, snowball sampling method was used. Since the purpose of the study is to explore correlations and not to study the prevalence of characteristics, a sample of approximately 125 subjects is sufficient for non-parametric correlation coefficients (with a significance of at least 0.28 and a power of 90% in two-tailed hypothesis testing and at an alpha significance level of 0.05) and the 95% confidence intervals for these coefficients.

### 2.3. Variables

The following variables were included to address the study objective and the initial hypothesis by establishing relationships between types of grief and psychosocial needs, interpersonal styles, and coping strategies:-Sociodemographic and health-related variables: Sex, age, level of education, nationality, chronic health problems, and prescribed drugs for mental health problems are the main variables used to describe the study participants.-Loss-related variables: The grieving situation is assessed by collecting data on the following variables: the degree of kinship with the deceased, cause of death, time elapsed since the loss occurred, and perceived current suffering due to the loss. As the main variable of interest for the study objectives, data on the type of grief exhibited by each participant were also collected: normal, at risk of complication, and complicated or maladapted grieving.-Interpersonal style: For assessing this aspect, the Spanish adaptation of the Interpersonal Adjective Scale (IAS) was used [10]. The IAS assesses the adjectives used to characterise interpersonal interactions, grouped into dimensions arising from the combination of responses to an eight-point Likert scale ranging from 1 (Extremely inaccurate) to 8 (Extremely accurate) [19]. Up to 64 adjectives are scored on the scale. The scoring of the various dimensions results in eight possible interpersonal styles, each graphically represented by an octant.-Coping strategies: The Spanish adaptation of Tobin’s Coping Strategies Inventory (CSI) [20] by Cano, Rodríguez, and García [11] was used to assess the coping strategies implemented by the participants. This is one of the most widely used instruments to measure coping. It has a factor structure composed of eight primary items, four secondary factors, and two tertiary factors. Responses are given on a five-point Likert scale ranging from 0 (Not at all) to 4 (Absolutely), depending on the degree of agreement. Four possible types of coping, classified as adequate, inadequate, problem-focused, and emotion-focused, can be identified based on the responses.-Psychosocial care needs: The Questionnaire for Psychosocial Nursing Diagnosis (QPSND) [12] was used to explore psychosocial care needs. The questionnaire was constructed and validated to assist in the identification of psychosocial nursing diagnoses using standardised nursing terminology. The total number of questions ranges from 36 to 61, depending on the answers provided by the patients. There are two types of responses: Likert-scale responses ranging from “Often” or “Always” to “Never” or “Not at all” (or vice versa) and dichotomous responses (“Yes” or “No”). There are ten question links, and the resulting combination of answers produces suggested outcomes in the form of up to 28 possible psychosocial diagnostic labels from the NANDA-I classification. These diagnostic labels are further grouped into six dimensions: Grief, Emotional/Self-Perception, Perception/Health Management, Behaviour/Social Interaction, Caregiver/Overload, and Body Image. In the Grief dimension, the QPSND proposes the following labels: *grief*, *risk for complicated grief*, and *complicated grief*. However, to remain consistent with the current naming of these diagnoses in the latest NANDA-I classification [15], the following labels will be used to report the results of this study: *readiness for enhanced grieving* (00285), *risk for maladaptive grieving* (00302), and *maladaptive grieving* (00301), which are equivalent to those resulting from the QPSND.

### 2.4. Data Collection Procedure

The data collected in the field notebooks were anonymised and entered into a database where the information recorded on paper was manually tabulated. Once this data transfer had been completed and the database had been cleaned for transcription errors, we carried out the planned tests to meet study objectives using descriptive and bivariate analyses. The data collection notebook contained a series of variables and instruments that are described below.

### 2.5. Data Analysis

The characteristics of the patients surveyed and the results of the instruments administered are summarised by expressing nominal variables as relative frequencies of their categories, expressing ordinal variables or variables that are not normally distributed as medians and percentiles (p5–p95 or interquartile ranges, IQRs), and expressing normally distributed variables as arithmetic means and standard deviations (SDs). To compare differences between variables, one variable was selected and compared with two or more groups using statistical tests that vary according to the nature of the variable: Pearson’s chi-squared test was used for nominal variables, the Mann–Whitney *U*-test was used for ordinal variables, and Student’s *t*-test was used for normal scalar variables. To analyse simple relationships and their association strength and to assign diagnostic labels, Pearson’s *r*, Kendall’s *tau*, and Mathews’ non-parametric *phi*, a symmetric contingency coefficient for nominal variables with two binary categories, were used. The prevalence of each diagnosis was provided along with the 95% confidence intervals. All tests were two-tailed and performed with an alpha significance of 0.05 using the SPSS v.25.0 statistical package for personal computers.

### 2.6. Ethical Considerations

The study was approved by the Research Ethics Committee at the Nuestra Señora de la Candelaria University Hospital Complex (CHUNSC_2020_22) located in the province of Santa Cruz de Tenerife (Canary Islands, Spain). All participants filled in the informed consent form once they had been briefed on the nature of the study, its characteristics, and the confidentiality of their data. Participants collaborated voluntarily and were free to withdraw from the study at any time without giving a reason.

Each participant was assigned a numerical code that was known only to the researchers for processing their data, thus ensuring the confidentiality and anonymity of the information collected from the participants, in compliance with the Spanish Organic Law 3/2018 of 5 December on Personal Data Protection and Guarantee of Digital Rights and adhering to the ethical principles of health research as set out in the Declaration of Helsinki.

## 3. Results

### 3.1. Sample Description

A total of 123 grieving individuals from the municipality of Güímar, Tenerife recruited between August 2014 and September 2019 participated in the study. Their mean age was 42.7 (±13.2) years, with women accounting for 64.2% (n = 79) of all participants. In terms of age ranges, 31.7% (n = 39) were between 18 and 34 years old, 31.7% (n = 39) were between 35 and 49 years old, 35% (n = 43) were between 50 and 64 years old, and 1.6% (n = 2) were aged 65 and over. Participants without education or with only primary education made up 8.1% (n = 10) of the sample, participants with secondary education or vocational training made up 48% (n = 59) of the sample, and those with university education represented 43.9% (n = 54) of the sample. Most participants (98.4%) were Spanish nationals. More than a third (34.1%; n = 42) had at least one chronic health condition, while 3.3% (n = 4) had been prescribed pharmacological treatment for a mental health problem. The most prevalent diseases included hypothyroidism (5.7%); allergies (4.9%); and diabetes mellitus, high blood pressure, and asthma (3.3% each).

Regarding the loss of a loved one, 86.2% (n = 106) reported that they were still suffering from their loss regardless of the time that had elapsed. The most frequent kinship with the deceased was that of a parent (35.8%), followed by that of a grandparent (26%). The causes of death most frequently reported by participants were cancer (27.5%), complications of chronic diseases (19.2%), cardiovascular problems (13.3%), and old age (12.5%). The time elapsed since their loss was one year or less for 53.7% of respondents, between one and two years for 21.1%, between two and five years for 10.6%, and more than five years for 14.6%.

The interpersonal styles resulting from the administration of the IAS were identified in different proportions across the eight possible styles: “Unassuming-Ingenuous” (26%), “Assured-Dominant” (24.4%), “Warm-Agreeable” (16.3%), “Gregarious-Extraverted” (9.8%), “Cold-Hearted” (9.8%), “Arrogant-Calculating” (6.5%), “Aloof-Introverted” (5.7%), and “Unassured-Submissive” (1.6%). Table 1 shows the score frequency for the 64 adjectives of the IAS. Combined scores indicate the predominant interpersonal style of each grieving person.

According to the CSI, 39% of the sample (n = 48) had an “adequate problem-focused coping” style, 35.8% (n = 44) had an “adequate emotion-focused coping” style; 15.4% (n = 19) had an “inadequate problem-focused coping” style, and 4.1% (n = 5) had an “inadequate emotion-focused coping” style. Additionally, 4.1% (n = 5) scored equally for “adequate problem-focused coping” and “adequate emotion-focused coping,” while the remaining 1.6% (n = 2) scored equally for “inadequate problem-focused coping” and “inadequate emotion-focused coping.” Therefore, regrouping the cases, 68.3% (n = 84) exhibited “adequate problem-focused coping” and 84.6% (n = 104) exhibited “adequate emotion-focused coping.” Summarising even further, 79.7% (n = 98) showed “adequate coping/management,” while the others showed “inadequate coping/management.”

The diagnostic labels for the psychosocial care needs identified after administering the QPSND are shown in Table 2. This table excludes grieving diagnoses, which were distributed in the sample of participants as follows: *readiness for enhanced grieving* (00285) (equivalent to a normal grieving process) was identified in 45.5% (n = 56) of participants, *risk for maladaptive grieving* (00302) was identified in 43.9% (n = 54) of participants, and *maladaptive grieving* (00301) was identified in 10.6% (n = 13) of participants.

The psychosocial diagnoses assigned by the QPSND are grouped into dimensions, which are found in the following proportions among the grieving participants: Emotional/Self-Perception = 77.2% (n = 95), Perception/Health Management = 46.3% (n = 57), Behaviour/Social Interaction = 39.8% (n = 49), Caregiver/Overload = 22.8% (n = 28), and Body Image = 1.6% (n = 2). It is important to note that 21.1% (n = 26) of the sample were caregivers of a relative or a dependent person.

### 3.2. Findings from Bivariate Analysis

#### 3.2.1. Relationship between the Type of Grieving and Sex

No significant differences were observed between sex and the type of grieving assigned by the QPSND; however, in the study population, females had a higher frequency of *risk for maladaptive grieving* and *maladaptive grieving* than males: 45.6% vs. 40.9% and 11.4% vs. 9.1%, respectively. Overall, women had a higher frequency of *chronic low self-esteem* than men, albeit at marginal significance: 16.5% vs. 4.5% (Pearson’s chi-squared = 3.744; *p* = 0.053).

A significantly higher number of psychosocial care needs (diagnoses assigned by the QPSND) were found among women who were assigned the *maladaptive grieving* diagnosis: 12 (IQR 10) vs. 4 (IQR 4); Mann–Whitney *U* = 104.000 (p = 0.001). Specifically, a higher frequency of the following psychosocial problems, also assigned by the QPSND, was observed among women with *maladaptive grieving* when compared to other grieving women:-*Moral distress*: 33.3% of women with *maladaptive grieving* vs. 7.1% of the other grieving women; chi-squared = 6.010; *p* = 0.044.-*Ineffective coping*: 77.8% vs. 14.3%; chi-squared = 19.036; *p* < 0.001.-*Impaired social interaction*: 33.3% vs. 2.9%; chi-squared = 12.494; *p* = 0.009.-*Social isolation*: 44.4% vs. 10.0%; chi-squared = 7.894; *p* = 0.019.-*Fear*: 77.8% vs. 37.1%; chi-squared = 5.414; *p* = 0.030.-*Chronic sorrow*: 33.3% vs. 2.9%; chi-squared = 12.494; *p* = 0.009.-*Hopelessness*: 55.6% vs. 1.4%; chi-squared = 33.291; *p* < 0.001.-*Powerlessness*: 66.7% vs. 17.1%; chi-squared = 11.117; *p* = 0.004.-*Chronic low self-esteem*: 55.6% vs. 11.4%; chi-squared = 11.295; *p* = 0.005.-*Loneliness*: 66.7% vs. 12.9%; chi-squared = 15.011; *p* = 0.001.-*Stress-anxiety syndrome*: 66.7% vs. 21.4%; chi-squared = 8.362; *p* = 0.009.-*Decreased diversional activity engagement in caregiver*s: 44.4% vs. 8.6%; chi-squared = 9.282; *p* = 0.013.

#### 3.2.2. Relationship between Types of Grieving and Level of Education

A higher frequency of *maladaptive grieving* was found in participants without education or with only primary education when compared to those with secondary education, vocational training, or university education: 40% vs. 8%; chi-squared = 9.975; *p* = 0.011.

#### 3.2.3. Relationship between Types of Grieving and Perceived Suffering

A higher frequency of *maladaptive grieving* was found among mourners who reported suffering from the loss of a loved one at the time of filling in the data collection notebook, although this difference was not significant: 11.3% vs. 5.9%.

#### 3.2.4. Relationship between Types of Grieving and Kinship

A higher frequency of *maladaptive grieving* was observed among participants who had lost a child compared with the rest of the participants, albeit with marginal statistical significance: 50% vs. 9.2%; chi-squared = 6.801; *p* = 0.055. Similarly, a higher prevalence of *risk for maladaptive grieving* was found among those who had experienced the loss of a partner: 100% vs. 42%; chi-squared = 5.283; *p* = 0.035.

#### 3.2.5. Relationship between Types of Grieving and Age

*Maladaptive grieving* was identified more frequently in the group of participants aged 50 and over, although the difference was not significant: 13.3% vs. 9.0%.

#### 3.2.6. Relationship between Types of Grieving and Coping Strategies

No significant associations were found between the CSI scores on coping and the presence of *maladaptive grieving* or *risk for maladaptive grieving*. However, a higher frequency of *maladaptive grieving* was found among participants who had been assigned the *ineffective coping* nursing diagnosis using the QPSND: 27.6% vs. 5.3%; chi-squared = 11.626; *p* = 0.002.

#### 3.2.7. Relationship between Types of Grieving and Interpersonal Styles

No significant relationships were observed between the prevalence of any of the eight possible personality styles from the IAS scale and the presence of the diagnoses *maladaptive grieving* or *risk for maladaptive grieving*. Conversely, when considering each of the 64 adjectives of the IAS tool and participants’ perceptions of how accurately each adjective defined their personality, ten adjectives were most frequently identified among those with *maladaptive grieving* and one adjective among those with *risk for maladaptive grieving* (Table 3).

#### 3.2.8. Other Relevant Associations

In keeping with the main purpose of this study, other associations of interest have been explored in relation to sociodemographic and health variables: those related to loss and types of grieving, interpersonal styles, and psychosocial care needs. As a result, no statistically significant associations were observed between the type of grieving and the presence of chronic health conditions, having drugs prescribed for mental health problems, the cause of death of the loved one, or the time elapsed since the loss. However, a higher frequency of *maladaptive grieving* was observed in cases where the death of the loved one had occurred one year or less before: 15.2% vs. 5.3%, without statistical significance.

Significant correlations were identified between the presence of *maladaptive grieving* and fourteen other psychosocial diagnoses resulting from the administration of the QPSND (Table 4).

No significant relationships were observed between each of the eight interpersonal styles of the IAS and the coping strategies in the CSI. However, associations were found between these strategies and fourteen interpersonal adjectives (Table 5).

## 4. Discussion

This study has identified a number of relationships between the types of grieving and the sociodemographic characteristics, psychosocial care needs, interpersonal styles, and coping strategies of a group of people who have experienced the loss of a loved one in a municipality on the island of Tenerife in the Canary Islands, Spain. Approximately one in three cases were experiencing a chronic health condition, with pharmacological treatment for mental health problems being very uncommon. Hypothyroidism was the most frequent condition and was predominant among women with a percentage close to the prevalence figures in Spain for this health problem [21].

A total of three out of four grieving people lost a loved one in the previous two years, and more than half of the grieving respondents in the sample had lost a loved one in the previous year. This could have a direct, logical, and expected impact on the identification of a wide range of psychosocial care needs. Time is known to be a key factor in the grieving experience, as it changes the perception and meaning of the loss, the narrative of the experience, the ties to the deceased, and one’s relationship to the world, which is completely transformed after such a disruptive event [22]. As a result, there is even a desynchronisation between the time elapsed for the grieving person and the ‘real’ time elapsed for the society around him or her.

### 4.1. Maladaptive Grieving

The 11th edition of the International Classification of Diseases (ICD-11) and the 5th edition of the Diagnostic and Statistical Manual of Mental Disorders (DSM-5) included Prolonged Grief Disorder as a diagnostic category alluding to a clinical picture in which problems associated with the grieving process extend over time causing dysfunction in people’s daily lives. These conditions generate intense grieving with worries about the deceased, feelings of emptiness, disinterest in life, and sleep problems. Furthermore, grieving complications have been shown to be correlated with acute coronary syndrome [1], with an estimated 7–10% of mourners not adjusting to their loss and developing complications as a result [23]. These figures are consistent with our study findings.

Perceived suffering caused by the loss of a loved one was present in almost nine out of ten cases, which would also be linked to the identification of psychosocial needs. Grief was classified as normal in almost half of the sample, while one in ten were labelled with maladaptive grieving and the remainder with risk for maladaptive grieving. These figures are similar to the distribution of these three categories among the sample of mourners included in the validation process of the QPSND itself [12]. Regarding impairment by dimension, almost eight out of ten participants were found to be impaired in the Emotional/Self-Perception dimension, followed by the Perception/Health Management dimension, which was found to be impaired in almost half of the sample, and Behaviour/Social Interaction, which was impaired in four out of ten cases. Problems in these areas are common during the grieving process and have been described by several authors [9,24,25]. The most frequent psychosocial labels (nonadherence to treatment, fear, risk for loneliness, risk for powerlessness, ineffective coping, and stress-anxiety syndrome) indicate emotional, coping, and social interaction problems more precisely. Risk for maladaptive grieving and maladaptive grieving showed higher percentages among women, albeit without statistical significance. A number of psychosocial problems were more frequently identified among women in the present study, with almost half of these problems also being significantly associated with being female in the QPSND validation study [12]: chronic sorrow, powerlessness, chronic low self-esteem, stress-anxiety syndrome, and decreased diversional activity engagement in caregivers. This suggests that women who have experienced the loss of a loved one may be more psychosocially vulnerable.

Level of education was another factor associated with the presence of maladaptive grieving, an aspect already known and reported by other researchers [26,27].

The most common kinship relationships were parents, followed by grandparents, and the most frequently reported causes of death were cancer and complications of chronic diseases. Regarding kinship, maladaptive grieving was more frequent among those who had lost a child, while risk for maladaptive grieving was more frequent among those who had lost a partner. The continuity of ties with deceased children allows parents to create very personal meanings, which are supplemented and reinforced within the cultural and social group to which they belong, thus shaping a lasting memory [28]. The lack of continued social and professional support along with the self-imposed isolation pursued by those who have experienced this type of loss may be more likely to trigger maladaptive grieving. Moreover, the loss of a partner is one of the most widely known life stressors, including within the context of the study population [29].

The nursing diagnosis of ineffective coping was found to be significantly associated with the presence of maladaptive grieving. One might ask whether it is coping with problems that leads to the occurrence of grieving complications or whether, on the contrary, a manifestation of maladaptive grieving itself makes it difficult to cope with problems. We did identify the aforementioned association between maladaptive grieving and ineffective coping, but not between maladaptive grieving and CSI scores. This could suggest that tools such as the QPSND which are based on standardised nursing languages should be used to assess coping and its relationship to the grieving process, rather than using other instruments less attuned to the nursing discipline, such as the CSI.

The relationship between the maladaptive grieving diagnosis and other psychosocial nursing diagnoses provides a clinical picture associated with the presence of this type of grieving, in which emotional, self-perception, and social problems are predominant. These other associated diagnoses are included in the defining characteristics of the NANDA-I diagnosis of maladaptive grieving, in particular anxiety, depressive symptoms, and expressing feeling detached from others.

### 4.2. Interpersonal Styles

In relation to interpersonal styles, “Unassuming-Ingenuous” was the most frequently identified, followed by “Assured-Dominant” and “Warm-Agreeable.” These self-reported interpersonal behavioural styles were also among the most frequent in the study testing the clinical properties of the IAS, both in the sub-sample of non-clinical and clinical subjects used by the authors [19]. The following adjectives most accurately defined their personalities and were present in virtually the entire sample: sympathetic, friendly, tender, and kind. Kindness as an interpersonal characteristic has been identified as a facilitator of community grieving through good social support [30]. In contrast, the adjectives most frequently reported by participants as inaccurate in defining their personality characteristics were: uncharitable, cruel, antisocial, and tricky.

The interpersonal adjectives associated with the presence of maladaptive grieving were: hard-hearted, dissocial, meek, introverted, not self-confident, domineering, antisocial, not self-assured, uncheery, and distant. These personality characteristics partly match the profile described by Felipe and Ávila [19], which is more prone to related clinical problems. These are interpersonal characteristics that are present in the NANDA-I classification. Thus, the nursing diagnosis of maladaptive grieving includes the following clinical manifestations as defining characteristics: expressing being overwhelmed, expressing feeling detached from others, expressing feeling of emptiness, expressing feeling stunned, expressing shock, and mistrust of others. Furthermore, an association was found between the interpersonal adjective unsympathetic and the nursing diagnosis of risk for maladaptive grieving, which includes attachment avoidance among its risk factors.

This study has some limitations. Firstly, the sample may be insufficient for its results to represent a broader profile of mourners in the study population, such as that of the municipality of Güímar (Tenerife). However, the objective of this study was not to describe a typical pattern or profile of grieving individuals in the study area at the population level, but to identify potential relationships between the psychosocial needs, personality styles, and coping strategies of a group of mourners. In addition, a potential sample selection bias could have been introduced due to the type of sampling method used. In future studies, a random selection of cases may be considered to address this issue. Nevertheless, the type of sampling used in this study was deemed to be the most relevant to recruit the number of participants previously estimated in accordance with the characteristics of our research.

## 5. Conclusions

This study presents the characteristics of a sample of grieving individuals from a specific municipality on the island of Tenerife in the Canary Islands, Spain. The types of grieving identified (normal, at-risk, maladaptive) are distributed in terms of frequency in a similar way to other samples used in comparable studies. The care needs identified are extensive and describe a profile of mourners with dysfunctions in the emotional, self-perception, behavioural, and social relations spheres. In addition, associations were found between a number of sociodemographic characteristics and the presence of *maladaptive grieving*, as well as between *maladaptive grieving* and certain interpersonal characteristics and also other psychosocial and, in particular, coping-related nursing diagnoses. Regarding the associations identified between coping strategies and the interpersonal characteristics of the mourners, we found that those with the best coping scores described themselves as self-confident, boastful, jovial, forceful, gentle-hearted, self-assured, outgoing, and/or neighbourly. By contrast, mourners who obtained poorer coping scores self-identified as shy, unsparkling, timid, unsociable, unbold, and/or bashful. Our findings provide a potential risk profile that nurses in the family and community settings could use to anticipate the assessment and identification of problems.

## Figures and Tables

**Table 1 healthcare-11-01244-t001:** Score frequency (%) on the Interpersonal Adjective Scale (IAS) responses for the sample of participants (n = 123).

Interpersonal Adjective		Extremely Inaccurate	Very Inaccurate	Quite Inaccurate	Slightly Inaccurate	Total: Inaccuracy	Slightly Accurate	Quite Accurate	Very Accurate	Extremely Accurate	Total: Accuracy
1	Assertive	0.8	5.7	2.4	4.9	13.8	22.0	37.4	19.5	7.3	86.2
2	Cunning	13.0	15.4	8.1	13.8	50.4	21.1	19.5	8.1	0.8	49.6
3	Hard-hearted	43.1	29.3	10.6	5.7	88.6	4.9	4.1	1.6	0.8	11.4
4	Dissocial	35.2	34.4	12.3	5.7	87.7	7.4	1.6	3.3	0.0	12.3
5	Meek	28.5	29.3	8.1	9.8	75.6	13.0	6.5	4.1	0.8	24.4
6	Undemanding	0.0	1.6	4.9	2.4	8.9	18.7	40.7	26.8	4.9	91.1
7	Accommodating	2.4	0.8	4.1	7.3	14.6	17.1	37.4	23.6	7.3	85.4
8	Perky	0.8	0.0	3.3	1.6	5.7	13.8	33.3	36.6	10.6	94.3
9	Dominant	10.6	15.4	8.9	19.5	54.5	25.2	10.6	9.8	0.0	45.5
10	Sly	14.8	11.5	8.2	16.4	50.8	27.9	12.3	5.7	3.3	48.8
11	Ruthless	61.0	20.3	8.1	3.3	92.7	2.4	1.6	3.3	0.0	7.3
12	Introverted	21.1	18.7	8.9	22.0	70.7	18.7	4.9	3.3	2.4	29.3
13	Shy	16.4	17.2	13.9	13.9	61.5	24.6	9.8	2.5	1.6	38.5
14	Uncrafty	6.5	4.9	6.5	8.9	26.8	11.4	27.6	27.6	6.5	73.2
15	Kind	0.0	0.8	1.6	0.8	3.3	7.4	32.8	41.8	14.8	96.7
16	Enthusiastic	0.8	0.0	2.4	7.3	10.6	12.2	29.3	35.8	12.2	89.4
17	Self-confident	1.6	4.1	4.9	9.8	20.5	19.7	35.2	19.7	4.9	79.5
18	Boastful	37.7	25.4	18.9	5.7	87.7	6.6	1.6	2.5	1.6	12.3
19	Uncharitable	35.0	40.7	14.6	5.7	95.9	1.6	0.8	0.8	0.8	4.1
20	Unsparkling	33.6	30.3	20.5	9.0	93.4	3.3	2.5	0.8	0.0	6.6
21	Soft-hearted	4.9	5.7	9.0	22.1	41.8	20.5	27.9	8.2	1.6	58.2
22	Uncalculating	0.8	0.8	0.8	3.3	5.7	9.0	31.0	35.2	18.9	94.3
23	Charitable	0.8	0.0	0.8	4.9	6.6	22.1	28.7	28.7	13.9	93.4
24	Jovial	0.0	1.7	1.7	8.3	11.6	15.7	32.2	33.1	7.4	88.4
25	Domineering	14.8	18.9	17.2	14.8	65.6	15.6	9.8	7.4	1.6	34.4
26	Crafty	2.5	4.9	4.9	7.4	19.7	25.4	35.2	13.9	5.7	80.3
27	Cruel	68.9	21.3	4.1	2.5	96.7	2.5	0.8	0.0	0.0	3.3
28	Antisocial	53.7	30.9	5.7	4.9	95.1	3.3	0.0	0.8	0.8	4.9
29	Unaggressive	5.7	2.4	0.8	13.8	22.7	7.3	35.0	29.3	5.7	77.3
30	Unsly	5.0	7.5	9.2	10.8	32.5	16.7	29.2	15.0	6.7	67.5
31	Tender-hearted	0.8	0.0	1.6	5.7	8.1	13.0	35.8	31.7	11.4	91.9
32	Extraverted	1.6	4.1	3.3	12.3	21.3	18.9	30.3	19.7	9.8	78.7
33	Forceful	3.3	6.6	4.1	9.8	23.8	13.9	36.1	18.0	8.2	76.2
34	Calculating	24.4	16.3	14.6	11.4	66.7	12.2	13.0	6.5	1.6	33.3
35	Iron-hearted	24.0	19.8	14.0	11.6	69.4	9.1	10.7	8.3	2.5	30.6
36	Unneighbourly	44.7	26.0	13.8	8.1	92.7	4.1	2.4	0.8	0.0	7.3
37	Timid	35.0	30.1	16.3	5.7	87.0	7.3	3.3	1.6	0.8	13.0
38	Boastless	4.9	4.9	5.7	8.1	23.6	20.3	30.1	21.1	4.9	76.4
39	Tender	0.0	1.6	0.8	0.8	3.3	9.8	44.7	32.5	9.8	96.7
40	Cheerful	0.8	0.0	0.8	4.1	5.7	18.0	40.2	29.5	6.6	94.3
41	Persistent	6.6	11.5	14.8	16.4	49.2	23.0	14.8	11.5	1.6	50.8
42	Tricky	52.8	34.1	7.3	3.3	97.6	0.8	0.8	0.0	0.8	2.4
43	Unsympathetic	30.9	35.0	15.4	8.9	90.2	8.1	0.8	0.8	0.0	9.8
44	Unsociable	38.2	35.8	8.9	4.1	87.0	8.9	2.4	0.8	0.8	13.0
45	Unbold	25.4	29.5	18.0	12.3	85.2	10.7	1.6	1.6	0.8	14.8
46	Unargumentative	2.5	2.5	4.9	7.4	17.2	15.6	38.5	21.3	7.4	82.8
47	Gentle-hearted	0.0	0.0	0.8	4.1	4.9	11.4	38.2	32.5	13.0	95.1
48	Friendly	0.0	0.0	0.8	2.4	3.4	4.9	39.0	36.6	16.3	96.8
49	Self-assured	1.6	4.1	3.3	11.4	20.3	20.3	33.3	17.9	8.1	79.7
50	Cocky	39.0	35.8	12.2	3.3	90.2	5.7	3.3	0.8	0.0	9.8
51	Warmthless	38.2	33.3	8.1	8.9	88.6	8.9	0.0	1.6	0.8	11.4
52	Uncheery	26.8	36.6	20.3	6.5	90.2	5.7	2.4	0.8	0.8	9.8
53	Bashful	13.1	17.2	13.1	19.7	63.1	23.8	4.9	5.7	2.5	36.9
54	Uncunning	0.8	1.6	4.9	4.1	11.4	19.5	35.0	23.6	10.6	88.6
55	Unauthoritative	0.0	1.7	1.7	7.4	10.7	15.7	33.9	28.1	11.6	89.3
56	Outgoing	0.8	1.6	1.6	7.3	11.4	17.9	35.0	23.6	12.2	88.6
57	Firm	0.8	7.4	7.4	12.3	27.9	19.7	27.9	20.5	4.1	72.1
58	Wily	31.7	17.9	6.5	5.7	61.8	17.1	13.0	6.5	1.6	38.2
59	Cold-hearted	47.2	22.0	11.4	8.9	89.4	5.7	2.4	1.6	0.8	10.6
60	Distant	41.8	27.0	11.5	9.0	89.3	7.4	3.3	0.0	0.0	10.7
61	Forceless	20.3	31.7	17.9	11.4	81.3	11.4	4.1	3.3	0.0	18.7
62	Unwily	10.6	17.9	12.2	17.9	58.5	14.6	10.6	13.0	3.3	41.5
63	Sympathetic	0.0	0.0	0.0	1.6	1.6	8.9	34.1	41.5	13.8	98.4
64	Neighbourly	0.0	0.8	2.4	4.9	8.1	11.4	27.6	34.1	18.7	91.9

**Table 2 healthcare-11-01244-t002:** Frequency of diagnostic proposals (%) after administration of the Questionnaire for Psychosocial Nursing Diagnosis (QPSND).

Psychosocial Diagnostic Label (QPSND)	Percentage (n)
Nonadherence to treatment	37.4 (46)
Fear	37.4 (46)
Risk for loneliness	27.6 (34)
Risk for powerlessness	26.8 (33)
Ineffective coping	23.6 (29)
Stress-anxiety syndrome	22.0 (27)
Powerlessness	19.5 (24)
Ineffective health maintenance	17.9 (22)
Stress overload	17.9 (22)
Ineffective medication self-management	17.1 (21)
Loneliness	15.4 (19)
Chronic low self-esteem	12.2 (15)
Social isolation	11.4 (14)
Risk for caregiver role strain	11.4 (14)
Decreased diversional activity engagement in caregivers	11.4 (14)
Caregiver role strain	9.8 (12)
Moral distress	7.3 (9)
Hopelessness	5.7 (7)
Impaired social interaction	4.9 (6)
Chronic sorrow	4.9 (6)
Spiritual distress	4.1 (5)
Risk for situational low self-esteem	1.6 (2)
Situational low self-esteem	1.6 (2)
Disturbed body image	1.6 (2)
Anxiety	0.8 (1)

**Table 3 healthcare-11-01244-t003:** Relationship between interpersonal adjectives and each of the following: maladaptive grieving and risk for maladaptive grieving.

Interpersonal Adjective on the Interpersonal Adjective Scale (IAS)		*Maladaptive Grieving*		Chi^2^
		No % (n)	Yes % (n)	
3. Hard-hearted	No	92.7 (101)	7.3 (8)	10.568
	Yes	64.3 (9)	35.7 (5)	
4. Dissocial	No	93.5 (100)	6.5 (7)	15.469
	Yes	60 (9)	40 (6)	
5. Meek	No	93.5 (87)	6.5 (6)	6.839
	Yes	76.7 (23)	23.3 (7)	
12. Introverted	No	93.1 (81)	6.9 (6)	4.242
	Yes	80.6 (29)	19.4 (7)	
17. Self-confident	No	76 (19)	24 (6)	5.881
	Yes	92.8 (90)	7.2 (7)	
25. Domineering	No	93.8 (75)	6.3 (5)	4.972
	Yes	80.5 (33)	19.5 (8)	
28. Antisocial	No	91.5 (107)	8.5 (10)	10.376
	Yes	50.0 (3)	50.0 (3)	
49. Self-assured	No	72.0 (18)	28.0 (7)	10.086
	Yes	93.9 (92)	6.1 (6)	
52. Uncheery	No	92.8 (103)	7.2 (8)	13.605
	Yes	58.3 (7)	41.7 (5)	
60. Distant	No	92.7 (101)	7.3 (8)	7.189
	Yes	69.2 (9)	30.8 (4)	
		** *Risk for Maladaptive Grieving* **		**Chi^2^**
		**No %(n)**	**Yes %(n)**	
43. Unsympathetic	No	60.4 (67)	39.6 (44)	8.395
	Yes	16.7 (2)	83.3 (10)	

**Table 4 healthcare-11-01244-t004:** Matching assignment between the psychosocial problems assigned by the Questionnaire for Psychosocial Nursing Diagnosis (QPSND) and maladaptive grieving.

Psychosocial Nursing Diagnoses Resulting from the Administration of the QPSND		*Maladaptive Grieving*		Phi
		No % (n)	Yes % (n)	
Spiritual distress	No	90.7 (107)	9.3 (11)	0.197
	Yes	60.0 (3)	40.0 (2)	
Moral distress	No	91.2 (104)	8.8 (10)	0.208
	Yes	66.7 (6)	33.3 (3)	
Ineffective coping	No	94.7 (89)	5.3 (5)	0.307
	Yes	72.4 (21)	27.6 (8)	
Impaired social interaction	No	91.5 (107)	8.5 (10)	0.290
	Yes	50.0 (3)	50.0 (3)	
Social isolation	No	92.7 (101)	7.3 (8)	0.293
	Yes	64.3 (9)	35.7 (5)	
Fear	No	94.8 (73)	5.2 (4)	0.226
	Yes	80.4 (37)	19.6 (9)	
Chronic sorrow	No	92.3 (108)	7.7 (9)	0.413
	Yes	33.3 (2)	66.7 (4)	
Hopelessness	No	94.0 (109)	6.0 (7)	0.600
	Yes	14.3 (1)	85.7 (6)	
Powerlessness	No	93.9 (93)	6.1 (6)	0.298
	Yes	70.8 (17)	29.2 (7)	
Chronic low self-esteem	No	93.5 (101)	6.5 (7)	0.357
	Yes	60.0 (9)	40.0 (6)	
Loneliness	No	94.2 (98)	5.8 (6)	0.365
	Yes	63.2 (12)	36.8 (7)	
Anxiety	No	90.2 (110)	9.8 (12)	0.263
	Yes	0.0 (0)	100.0 (1)	
Stress overload	No	93.8 (90)	6.3 (6)	0.265
	Yes	74.1 (20)	25.9 (7)	
Decreased diversional activity engagement	No	91.7 (100)	8.3 (9)	0.210
	Yes	71.4 (10)	28.6 (4)	

**Table 5 healthcare-11-01244-t005:** Relationship between interpersonal adjectives and coping/problem management according to the Coping Strategies Inventory (CSI).

Interpersonal Adjective on the Interpersonal Adjective Scale (IAS)		*Adequate Problem-Focused Coping*		Chi^2^	*p*-Value	*Adequate Emotion-Focused Coping*		Chi^2^	*p*-Value	*Overall Adequate Coping*		Chi^2^	*p*-Value
		No % (n)	Yes % (n)			No % (n)	Yes % (n)			No % (n)	Yes % (n)		
13. Shy	No	24 (18)	76 (57)	3.758	0.053	8.3 (6)	91.7 (66)	4.094	0.043				
	Yes	40.9 (18)	59.1 (26)			21.3 (10)	78.7 (37)						
17. Self-confident	No	52.2 (12)	47.8 (11)	6.493	0.011	33.3 (8)	66.7 (16)	10.218	0.004	44.0 (11)	56.0 (14)	10.665	0.001
	Yes	25.0 (24)	75.0 (72)			8.4 (8)	91.6 (87)			14.4 (14)	85.6 (83)		
18. Boastful	No									23.4 (25)	76.6 (82)	4.408	0.039
	Yes									0.0 (0)	100.0 (15)		
20. Unsparkling	No					11.6 (13)	88.4 (99)	5.528	0.050				
	Yes					42.9 (3)	57.1 (4)						
24. Jovial	No					42.9 (6)	57.1 (8)	11.633	0.004	42.9 (6)	57.1 (8)	4.758	0.040
	Yes					9.6 (10)	90.4 (94)			17.8 (19)	82.2 (88)		
33. Forceful	No	48.3 (14)	51.7 (15)	6.572	0.010	33.3 (9)	66.7 (18)	11.870	0.002	41.4 (12)	58.6 (17)	10.187	0.001
	Yes	23.3 (21)	76.7 (69)			7.6 (7)	92.4 (85)			14.0 (13)	86.0 (80)		
37. Timid	No	26.0 (17)	74.0 (77)	6.058	0.020								
	Yes	56.3 (9)	43.8 (7)										
44. Unsociable	No	23.1 (24)	76.9 (80)	17.802	<0.001	9.6 (10)	90.4 (94)	9.331	0.008	15.0 (16)	85.0 (91)	14.658	0.001
	Yes	75.0 (12)	25.0 (4)			37.5 (6)	62.5 (10)			56.3 (9)	43.8 (7)		
45. Unbold	No	24.8 (25)	75.2 (76)	9.571	0.002	10.8 (11)	89.2 (91)	4.345	0.053	15.4 (16)	84.6 (88)	11.285	0.002
	Yes	61.1 (11)	38.9 (7)			29.4 (5)	70.6 (12)			50.0 (9)	50.0 (9)		
47. Gentle-hearted	No									66.7 (4)	33.3 (2)	8.365	0.015
	Yes									17.9 (21)	82.1 (96)		
49. Self-assured	No	52.2 (12)	47.8 (11)	6.662	0.010	34.8 (8)	65.2 (15)	11.328	0.003	44.0 (11)	56.0 (14)	10.860	0.001
	Yes	24.7 (24)	75.3 (73)			8.2 (8)	91.8 (89)			14.3 (14)	85.7 (84)		
53. Bashful	No	22.7 (17)	77.3 (58)	5.531	0.019								
	Yes	43.2 (19)	56.8 (25)										
56. Outgoing	No	57.1 (8)	42.9 (6)	5.560	0.028	46.2 (6)	53.8 (7)	13.591	0.002	50.0 (7)	50.0 (7)	8.591	0.008
	Yes	26.4 (28)	73.6 (78)			9.3(10)	90.7 (97)			16.5 (18)	83.5 (91)		
64. Neighbourly	No					40.0 (4)	60.0 (6)	6.713	0.028	50.0 (5)	50.0 (5)	5.919	0.029
	Yes					10.9 (12)	89.1 (98)			17.7 (20)	82.3 (93)		

## Data Availability

The data presented in this study are available upon request from the corresponding author. The data are not publicly available due to privacy/ethical restrictions.

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
