# Peer review of "Psychosocial Care Needs, Personality Styles, and Coping Strategies of Mourners in a Rural Municipality in Spain: An Observational Study"

_healthcare, 2023, doi:10.3390/healthcare11091244_

Round 1

Reviewer 1 Report

/C/V

Author Response

Our research team has studied each of the suggestions made by the reviewers and has made the appropriate modifications.
Please see the attached file.

Reviewer 2 Report

This article consists in an observational study, reporting relationships between interpersonal styles, coping strategies, and psychosocial care needs in a sample of 123 grieving people.

Introduction: 

1)  The authors should better introduce the objective of their work. The introduction is interesting but needs to allow the reader to understand why the authors conducted the study and its implications. 

The introduction should guide the reader in understanding why the proposed study is relevant, not just give an overview of the topic of grieving.

2) "As such, we believe that further insight into different types of grief could be obtained by describing the interpersonal characteristics and coping styles of grieving people living in a rural setting while considering their individual psychosocial needs."

The authors should specify better 1) their aim (why a rural setting? what are the differences in grieving in rural vs urban populations?), 2) the relevance of the study (what are the benefit of a better characterization of the grieving?)

3) The authors should introduce specific hypotheses on the variables collected (2.3 in methods): why did they collect these data? Why are these data relevant for grieving?

Materials and Methods:

1) although authors described grieving as a reaction to several situations (e.g., job loss or the end of a relationship; Introduction), they focused only on participants that have lost a loved one; why?

2) table 1, table 4 and table 5 are very confusing

3) probably the authors should present their data with figures 

3) "however, in the study population, females had a higher frequency of Risk for maladaptive grieving and Maladaptive grieving than males: 45.6% vs 40.9% and 11.4% vs 9.1% respectively."

are all these quantitative considerations? If yes, the authors should report the statistics.

4) "Specifically, a higher frequency of the following psychosocial problems, also assigned by the QPSND, was observed among women with Maladaptive grieving when compared to other grieving 256 women."

The authors performed many analyses. Are these comparisons corrected for multiple comparisons (e.g., Bonferroni or Bonferroni-Holm)?

5) In table 3, "Risk for maladaptive grieving" should be in bold

Discussion: 

1) The discussion could be shorter and more precise. What is the take-home message? Which are the implications of these results?

Authors should make the discussion clearer, for instance, by dividing it into sub-sections. Furthermore, they should not merely repeat their results but try to provide a general meaning to the study. 

General comment: 

1) The manuscript presents revision (see discussion); the authors should provide a final version without visible corrections

Abstract: 

1) The abstract does not report any take-home message. The authors should reframe the abstract in the final part (from line 23 to line 34). 

At the moment, the abstract reported a description of the sample size, without specifications about the relevance of these analyses, hyphotheses, take-home message and the implication of the present study.

Author Response

(The authors gave the same response as above.)

Reviewer 3 Report

I read the article with great interest. 

Please consider correcting the following.

The layout of Table 1 is difficult to read.

For example, terms such as "Extremely inaccurate" and "Very inaccurate" are arranged vertically.

I believe this needs to be corrected to avoid confusion among readers.

The layout of Table 2 differs from Table 1.

Table 2 has horizontal ruled lines.

The layout needs to be unified. 

I believe that the results of the statistical analysis on page 9, lines 246-270, would be better understood by the reader if they were tabulated.

Author Response

(The authors gave the same response as above.)

Round 2

Reviewer 2 Report

the article is now suitable for publication